# A Raman lidar at Maïdo Observatory (Reunion Island) to measure water vapor in the troposphere and lower stratosphere: calibration and validation

Hélène Vérèmes<sup>(1,2)</sup>, Guillaume Payen<sup>2</sup>, Philippe Keckhut<sup>3</sup>, Valentin Duflot<sup>(1,2)</sup>, Jean-Luc Baray<sup>4</sup>, Jean-

5 Pierre Cammas<sup>(1,2)</sup>, Jimmy Leclair de Bellevue<sup>1</sup>, Stéphanie Evan<sup>1</sup>, Françoise Posny<sup>(1,2)</sup>, Franck Gabarrot<sup>2</sup>, Jean-Marc Metzger<sup>2</sup>, Nicolas Marquestaut<sup>2</sup>, Susanne Meier<sup>5</sup>, Holger Vömel<sup>6</sup> and Ruud Dirksen<sup>5</sup>

<sup>1</sup>Laboratoire de l'Atmosphère et des Cyclones, UMR8105, Saint-Denis de La Réunion, France <sup>2</sup>Observatoire des Sciences de l'Univers de La Réunion, UMS3365, Saint-Denis de la Réunion, France

- <sup>3</sup>Laboratoire ATmosphères, Milieux, Observations Spatiales-IPSL, UMR8190 CNRS, UVSQ, UPMC, Guyancourt, France
   <sup>4</sup>Laboratoire de Météorologie Physique, UMR6016, Observatoire de Physique du Globe de Clermont-Ferrand, Université Clermont Auvergne, Clermont-Ferrand, France
   <sup>5</sup>Deutscher Wetterdienst, Meteorological Observatory Lindenberg, Lindenberg, Germany
- <sup>6</sup>National Center for Atmospheric Research, Boulder, CO, USA

Correspondence to: Hélène Vérèmes (helene.veremes@univ-reunion.fr)

Abstract. The Maïdo high-altitude observatory located in Reunion Island (21°S, 55.5°E) is equipped with Lidar1200, an innovative Raman lidar designed to measure the water vapor mixing ratio in the troposphere and the lower stratosphere. The

- calibration methodology is based on a GNSS (Global Navigation Satellite System) IWV (Integrated Water Vapor) dataset and lamp measurements. The mean relative standard error on the calibration coefficient is around 2.7%. Two years of lidar water vapor measurements from November 2013 to October 2015 are now processed. By comparing CFH (Cryogenic Frost point Hygrometer) radiosonde profiles with the Raman lidar profiles, the ability of the lidar to provide accurate measurements is possible up to 22 km. The ability of measuring water vapor mixing ratios of a few ppmv in the lower stratosphere is
- demonstrated with a 48-hours integration time period, an absolute error lower than 0.8 ppmv and a relative error less than 20%. This Raman lidar is dedicated to provide regular profiles of water vapor measurements with a high vertical resolution and low uncertainties to international networks; in the wider interest of research on stratosphere-troposphere exchange processes and on the long-term survey of water vapor in the upper troposphere and lower stratosphere in the Southern Hemisphere. A strategy of data sampling and filtering is proposed to meet these objectives with regard to the altitude range requested. 10-min time
- integration and 65-90 m vertical resolution ensure a vertical profile reaching 10 km, but more than 2800 minutes and a vertical resolution of 150-1300 m are necessary to reach the lower stratosphere with an uncertainty less than 20%.

# 1. 1 Introduction

To monitor potential climate changes, observations of essential climate variables (ECV) such as atmospheric water vapor (Bojinski et al., 2014) are necessary. Long-term series allow the international community to progress on important

climatological issues, such as the contributions of stratospheric water vapor to decadal changes in the rate of global warming (Solomon et al., 2010). Selection criteria are important for the certification of networks. These criteria include: long-term stability and regularity. When supplying the databases, very precise and detailed metadata files of the physical quantities have to be provided. And most importantly, an uncertainty has to be associated to the data. This last exercise is realized through the examination of the data processing algorithms and calibration methodologies. This rigor is essential to monitor efficiently the

atmosphere and climate changes (Immler et al., 2010). One of the challenging ECV to measure is water vapor mainly in the upper troposphere and lower stratosphere (GCOS, 2003). Water vapor is the main greenhouse gas. The factors influencing its spatio-temporal variability are various: convection, precipitations, temperature (Kennett and Toumi, 2005), transport and dynamical processes from eddies to synoptic scale events. The residence time of the water vapor in the atmosphere is several

- 5 days to weeks (Trenberth, 1998; Läderach and Sodemann, 2016). Nevertheless, its temporal variability can be high at a scale of dozens of minutes, and its spatial variability can be less than one kilometer (Vogelmann et al., 2011, 2015). To track, monitor and set water vapor trends in the troposphere and in the lower stratosphere, it is essential that dense and regular measurements supply global networks as NDACC (Network for the Detection of Atmospheric Composition Change; Kurylo and Solomon, 1990) and GRUAN (GCOS Reference Upper Air Network; Seidel et al., 2009, Bodeker et al., 2015). The
- NDACC community gives a stronger focus on the study of the dynamics and on chemical processes of the UT/LS (Upper Troposphere/Lower Stratosphere) that represents a key area of exchanges between the two main layers. The characterization of the water vapor in the troposphere and more specifically in the UT/LS is one of the scientific goals of GRUAN (Seidel et al., 2009). This network ensures the quality and the density of stations in order to face this scientific issue. It is not sufficient to ensure a global covering and frequent measurements, it is also essential to trust measurements quality and accuracy up to
- the lower stratosphere.

Water vapor measurement techniques are various: in situ or remote sensing, ground based, airborne or space based (Wulfmeyer et al., 2015). A limited number of instruments is able to accurately measure the vertical distribution of the water vapor in the UT/LS. The sondes with a capacitive sensor are not reliable enough in the stratosphere (WMO, 2011). Few of these sondes measure water vapor in the subtropical upper troposphere. Among the radiosondes, the hygrometers are the most

- efficient. The CFH (Cryogenic Frost point Hygrometer) exhibits accuracy better than 4% in the tropical lower stratosphere, to 9% around the tropopause and about 10% at 28 km (Vömel et al., 2007). They are shown to be good in the UT/LS (Vömel et al., 2007). Nevertheless, the CFH are rarely launched on a routine basis at these stations mainly because of their cost. GRUAN recommends to launch at least one radiosounding per month with a CFH or similar sondes (Bodeker et al., 2015). Regarding global covering satellites measurements, the MLS (Microwave Limb Sounder) is very accurate (Read et al., 2007) and even
- the most efficient in the lower stratosphere. Between 100 and 121 hPa, the accuracy (bias) of the MLS is around 15-20% (8-12%). At 83 hPa, it is reduced to 10% (7%). The detection threshold is extremely low: 0.1 ppmv above 150 hPa (Livesey et al., 2011). Few of the others on board satellite instruments, in operation today, can measure in the vicinity of the tropopause and even less the lower stratosphere. If the accuracy of MLS is very high, its horizontal resolution is poor and its vertical resolution is of the order of 2-3 km. Other remote sensing measurements (ground-based or on board instruments) are based on
- the lidar (LIght Detection And Ranging) systems. Water vapor profiles can also be measured by two lidar techniques: DIAL (DIfferential Absorption Lidar) and Raman lidar. The overview of the capacities of the ground-based DIAL lidar by Wulfmeyer et al. (2015) reports that they have a range of 10 km (for night measurements) with a vertical resolution from 15 to 900 m and an accuracy of 1 to 10% (depending on the altitude). No calibration factor is needed, avoiding a supplementary uncertainty. Nevertheless, the DIAL measurement is limited in altitude because of the large absorption in the lower troposphere in case of
- high water vapor (Turner et al., 2000).

In opposite, some of the NDACC Raman lidars (all ground-based) are able to reach the UT/LS as the Purple Crow Lidar (Canada; Argall et al., 2007), the Table Mountain Facility (United-States; Leblanc et al., 2012) and Tor Vergata system (Italy) (Dionisi et al., 2010) lidars. Nevertheless, the altitude of the tropopause is lower in such mid-latitude areas than in the (sub)tropics. The Mauna Loa Observatory Raman lidar is able to reach 15 km on a routine basis (Barnes et al., 2008). Raman

lidar could be able to measure water vapor in the tropical UT/LS but such performances remain to be demonstrated and quantified precisely. The calibration of the data of the Raman lidars is important for the stability of the measurement on a short and long-term basis (Sherlock et al., 1999a) and will induce an uncertainty of at least 5% (Wandiger et al., 2005) no matter the complementary measurement used. Calibration is typically done using water vapor profiles from radiosoundings (Sherlock et et al., 2005) and will induce an uncertainty of a least 5% (Wandiger et al., 2005) no matter the complementary measurement used. Calibration is typically done using water vapor profiles from radiosoundings (Sherlock et et al., 2005) no matter the complementary measurement used.

al., 1999a). All the Raman water vapor lidars of the NDACC used Vaisala sondes to calibrate their database. Different measurements have been tested and intercompared to calibrate the data (GNSS, radiometer and optimized matching methods when using radiosoundings) during different campaigns: DéméVap (Développements Méthodologiques pour le sondage de la Vapeur d'eau dans l'atmosphère; Bock et al., 2013), MOHAVE (Measurements Of Humidity in the Atmosphere and Validation

- Experiments; Leblanc et al., 2012), and HOPE (HD(CP)<sup>2</sup> Observational Prototype Experiment; Foth et al., 2015). The uncertainty on the calibration factor calculated by the different instruments was between 5 to 10%. One of the conclusions of these studies is that the GNSS IWV (Integrated Water Vapor) is appropriate to calibrate the Raman lidar, the resulting uncertainty being around 7% during DéMéVap (Bock et al., 2013). The key point to use the GNSS IWV for the calibration of Raman lidar water vapor measurements is the lidar's ability to retrieve the total water vapor column to ensure that the same
- columns are probed by the 2 instruments; in other words: whether or not the lidar is able to measure water vapor close enough to the ground.

In Reunion Island, a new Raman lidar has been designed to monitor simultaneously the water vapor from the ground up to the lower stratosphere and the temperature in the stratosphere and the mesosphere. The system is an updated version of the former Rayleigh-Mie-Raman system in operation at Saint-Denis (Reunion Island) between 2002 and 2010. Some critical

- points have been addressed in the upgrade, including fluorescence, power and parallax effects, in order to optimize the configuration of the system (Hoareau et al., 2012; Sherlock et al., 1999b). In October 2012, this new lidar system has been set up at a higher altitude: the Maïdo facility located at 2160 m asl (Baray et al., 2013). At the same time, most of the instruments of the OPAR (Observatoire de Physique de l'Atmosphère de la Réunion) moved to the observatory. In order to optimize the instrumental configuration and to evaluate a first set of data, two measurement campaigns were organized: MALICCA-1
- (MAïdo LIdar Calibration CAmpaign) in April 2013 (Keckhut et al., 2015) and MALICCA-2 in November 2013. The MALICCA-1 dataset shows a good agreement between the water vapor data measured by the lidar and those measured by the other instruments (Vaisala RS92, MLS, Dionisi et al., 2015). The relative difference between the lidar profiles and 15 simultaneous radiosoundings (Vaisala RS92) was lower than 10% for the lower and middle troposphere and between 10 and 20% for the upper troposphere. Two integration methods have been tested: 240 minutes with an uncertainty of 2 ppmv between
- 17 and 20 km, and a monthly integration (i.e. an integration of ~ 1920 min) with an uncertainty of 1 ppmv at 20 km, which demonstrates the ability of the lidar to measure quantities of only few ppmv in the UT/LS (Dionisi et al., 2015). The monthly mean profile of water vapor based on MLS data agrees well with the mean lidar profile of MALICCA-1 in the lower stratosphere. The main conclusions of Dionisi et al. (2015), based on two weeks of intensive measurements during a campaign, need to be extended on a longer 2-years period of routine measurement and to be reviewed. GNSS data can be used to calibrate
- the water vapor mixing ratio profiles but it is still necessary to develop a robust methodology of calibration feasible on long term, as required for participation in international networks such as NDACC. It is noteworthy that this Maïdo Raman water vapor lidar (called hereafter "Lidar1200") was recently provisionally affiliated within the NDACC. The conclusive affiliation occurs when absence of fluorescence and a stable calibration method are both demonstrated using validation campaigns involving frost-point hygrometer measurements.
- The main objectives of this paper are to assess the ability of this lidar to monitor the water vapor in the lower tropical stratosphere on a long-term basis and to introduce its 2-years dataset of water vapor profiles. We will describe the instrument, the data processing and the calibration methodology. A routine calibration methodology, based on GNSS observations, has been developed in order to improve the reliability of the calibration coefficient and the robustness of the measurements. This methodology is detailed in Sect. 2. The validation of the profiles by comparing the lidar data with CFH radiosoundings will
- be presented in the Sect. 3. The evaluation of the performances of the instrument is detailed in Sect. 4 from fine scale structures detection to the potential establishment of climatologies and then trends in the UT/LS. Finally, an overview of the 2-years dataset is given (Sect. 5).

#### 2. Description of the system, the data processing and the calibration methodology

#### 2.1 Routine Raman system measurements

Before being transferred to the Maïdo Observatory in 2012, the Raman water vapor lidar operated in the north of Reunion Island, at Saint-Denis, at sea level (Hoareau et al., 2012). The system was upgraded to work at 355 nm, a more efficient

- wavelength than 532 nm (Dionisi et al., 2015). Laser pulses are generated by two Quanta Ray Nd:Yag lasers with a repetition rate of 30 Hz. The use of two lasers increases significantly the power of the signal. They are synchronized through a pulse generator cube with an uncertainty less than 20 ns. The emitting pulse of each laser has an energy of 375 mJ.pulse<sup>-1</sup> and a duration of 9 ns and they are gathered through a polarization cube. The geometry for transmitter and receiver is coaxial for three different reasons: i) to avoid parallax effects, ii) to extend the measurements down to the ground and iii) to facilitate the
- alignment. The backscattered signal is collected by a Newtonian telescope with a primary mirror of 1.2 m diameter. When collecting small Raman scattering by water vapor compared with the large elastic scattering able to generate fluorescence, the use of an optical fiber is an important issue. It has been shown that the fluorescence in such cables can cause systematic biases (Sherlock et al., 1999b). Thus, no optical fiber is used. A spectrometer is used directly after this telescope to separate the Raman and Rayleigh signals. Thanks to a diaphragm field stop at the entrance of the spectrometer, the field of view (FOV) of
- the system is adjustable (from 3.0 to 0.5 mrad). On a routine basis, a 2 mm FOV (0.5 mrad) allows the reduction of the background light, limits the photo-counting saturation from low altitude scattering. The overlap factor is identical for both channels. Thus, water vapor profiles can be retrieved down to the ground. Details on the different tests realized during MALICCA that led to the final choice of the operational optical configuration are given in Dionisi et al. (2015). The spectrometer is composed of dichroic beam splitters and interference filters which separate the backscattered light. Hamamatsu
- miniature PMT and Licel transient recorders are used for the photo detection and data acquisition in photon counting.

#### 2.2 Water vapor data processing

The initial data processing algorithm (Dionisi et al., 2015) has been upgraded to version 2.3.1 to include the identification and quantification of the uncertainties and the vertical resolution determination.

# 2.2.1 Identification and quantification of the uncertainties

A full analysis of identification of the sources and quantification of the different uncertainties has been carried out. The total absolute error ( $\Delta$ WVMR) on the calibrated profiles (WVMR) is calculated as followed:

$$\frac{\Delta WVMR}{WVMR} = \sqrt{\left(\frac{dw}{w}\right)^2 + \left(\frac{dC}{C}\right)^2 + \left(\frac{d\Gamma}{\Gamma}\right)^2} \tag{1}$$

where  $\left(\frac{dw}{w}\right)$  is the relative error associated with the statistical error of the detectors (PMT),  $\left(\frac{dC}{C}\right)$  is the uncertainty due to the

- calibration coefficient and  $\left(\frac{d\Gamma}{\Gamma}\right)$  is due to the extinction coefficient term. The different sources of the assumed uncorrelated statistical error are associated with the counting of the number of photons collected by the detector for the water vapor and nitrogen channels depending on the altitude. The sources of systematic errors are the determination of the calibration coefficient, and of the cross-sections at the wavelengths of the water vapor and nitrogen Raman channels. Finally, the molecular extinction/density can be retrieved by either a model, a climatology or measurements with a good accuracy regarding the
- relatively weak variability of the atmospheric density at a given level and is based on the pressure and temperature profiles. For the Lidar1200 data, we identify that the main uncertainties are:
  - the uncertainty on the model of the density profile (extracted from a climatology) that has been arbitrarily fixed to

15%. After propagation of the error, it represents a negligible uncertainty of only 0.01 to 0.02% on the data. We suppose that the uncertainty due to aerosols on the factor of transmission is negligible compared to the uncertainty on the molecular transmission. Indeed, the effect of aerosols on Raman channels in the UV is low.

- the uncertainty on the calibration ranges from 1.6% to 3.6% for the 2013-2015 period. The mean standard error is about 2.7%. This will be developed in Sect. 2.3. It is independent of the altitude.
- the statistical error has the more important contribution, it increases with the altitude and depends on different characteristics of the instrument and the signal varying from one measurement to another (integration time, background). This error will drive the reliability of the data in the UT/LS.

The total error is directly influenced by the calibration and the quality of the measurements. It varies because of the statistical

and systematic errors in the lower troposphere. In the middle troposphere up to the lower stratosphere, the statistical errors determine the reliability of the profiles. The uncertainty depends strongly on the integration time and the filtering of the signal. Thus, it is important to use a suitable filter regarding the vertical resolution and the order of magnitude of the total error.

#### 2.2.2 Vertical resolution

The vertical resolution of the raw data is 15 m. Data are smoothed with a filter using the Blackman coefficients: thus, the final 15 resolution is different from the initial resolution. The number of points of the filter varies with the altitude, the number increasing with altitude to compensate the signal to noise ratio (SNR) decrease. The NDACC has formulated and adopted two standardized definitions for the calculation of the final vertical resolution for its lidars based on 1) cut-off frequency of digital filters (used here) or 2) the full-width at half-maximum of a finite impulse response (Leblanc et al., 2016). Based on the number of points used for the filter to vertically average the data, the vertical resolutions (Fig. 1) that can be derived from these

20 definitions are 100-200 m in the lowest layers, 500 m in the mid-troposphere, 600 m in the upper troposphere and 700-750 m in the lower stratosphere, for a filter using the Blackman coefficients reaching 121 points at 20 km asl.

To summarize, the uncalibrated profiles are processed with a vertical resolution, and a time integration depending on the water vapor variability at several levels. In order to convert the backscattered radiation profiles into water vapor mixing ratio profiles, it is necessary to calculate afterward a calibration coefficient from water vapor column ancillary data. The

25 specific calibration methodology that has been developed for the Lidar1200 is described in the next subsection. The validation and evaluation of the performances of the lidar is detailed in Sect. 3 and 4.

#### 2.3 Calibration methodology

#### 2.3.1 The GNSS technique

The GPS and GLONASS (GLObal NAvigation Satellite System) satellite constellations signals are collected by a ground-30 based receiver. With respect to propagation in a vacuum, the signal travelling between a GPS satellite (altitude of 20,200 km) and a ground-based receiver is delayed by atmospheric constituents (dry air, and water vapor). To determine the IWV from GNSS data, the total atmospheric zenithal delay (ZTD) is estimated. Using surface pressure information, the ZTD can be divided into a hydrostatic term, i.e. the Zenithal Hydrostatic delay (ZHD or so-called dry delay) calculated through the Saastamoinen formula (Saastamoinen, 1972). The hydrostatic delay, which is derived by applying the condition that

- hydrostatic equilibrium is satisfied, depends on the total weight of the atmosphere above. The difference term accounts for the Zenithal Wet Delay (ZWD, so-called wet delay). The wet delay is the propagation delay experienced by GPS signals due mainly to water vapor abundance. ZWD is converted into IWV, using surface temperature and empirical formulas (Bevis et al., 1992; Emardson and Derks, 2000). The raw data inversion process and its uncertainty is mainly controlled by mapping the Slant Total Delay (STD, in any direction) to the delay at zenith to which horizontal gradients (North-South and East-West) are
- added.

Many authors assessed the accuracy in GPS IWV, based on comparisons with radiosondes, sun photometers,

microwave radiometers, lidars and interferometers (Foelsche and Kirchengast, 2001; Niell et al., 2001; Bock et al., 2016). These techniques agree for about 1-2 kg.m<sup>-2</sup> for typical values between 5 and 30 kg.m<sup>-2</sup>. In practical, an accuracy ranging from 0.5 to 2.5 kg.m<sup>-2</sup> have been observed regarding the localization of the measurements but few sources of uncertainties still need to be investigated (Bock et al., 2013).

A TRIMBLE-NETR9 receiver (MAIG) has been installed at the Maïdo Observatory in 2013 in order to receive the GNSS data (see <u>http://rgp.ign.fr/STATIONS/#MAIG</u>). The GNSS network used to calculate IWV above the station includes 20 other local stations disseminated over the island, and in order to ensure efficiently high numbers of baselines for the differential data processing through GAMIT software v10.32 (King and Bock, 2007), 15 overseas stations in the Indian Ocean are included. The typical cutoff elevation angle is fixed to 10° to ensure the most local sounded area of water vapor as possible.

Currently, the database produces hourly IWV, 24 hours at the Observatory elevation. The precision is in agreement with the literature and is about 1 to 2 mm. The complete evaluation of the uncertainty will be further detailed in a future publication.

# 2.3.2 Relevance of the use of GNSS IWV for calibration

The raw data of the Lidar1200 need to be calibrated. Two ancillary measurements might be used: radiosondes or GNSS. The main concern for the calibration is to be able to have collocated and simultaneous measurements on a routine basis. We consider

- that space-time criteria for colocation do not match between the biweekly normal operation of the Raman lidar (15:00-21:00 UTC) at the Maïdo Observatory and the daily 12:00 UTC radiosounding (Météo-France) or the weekly 10:00 UTC ozone radiosounding (NDACC/SHADOZ OPAR site) based at the airport which is 20 km from the Maïdo Observatory. Launching two sondes per week from the Maïdo Observatory would have a significant financial and logistical cost. These are the main reasons in choosing the GNSS for the calibration. GNSS measurements are abundant, collocated and simultaneous to the lidar
- data. IWV is calculated each hour providing that the uncertainty of the estimated ZTD by GAMIT remains below the standard threshold of 30 mm which occurs around 95 % of the time. The other reasons are the independence of the lidar series regarding the radiosoundings series that have revealed some discontinuities in the past due to the relative humidity sensor calibration and the hourly basis availability of the GNSS data. Since 2013, the lidar profiles can be calibrated using integrated water vapor columns obtained from these GNSS measurements.
- As stated earlier, the main prerequisite to the use of GNSS IWV to calibrate Raman lidar water vapor measurements is the ability of the lidar to actually probe the IWV, i.e. to start measurements at the ground level. The emission and reception parts being coaxial, the full recovering altitude of the lidar is very low. The recovering of the laser beam with the field of view of the telescope is partial from the ground (i.e. 2.2 km asl) to 4 km asl. The data processing is based on a ratio between both channels (H<sub>2</sub>O and N<sub>2</sub>), there is no correction associated to the recovering. The first available point of the lidar water vapor
- profile is 15 m above the instrument (vertical resolution of the raw data). This somehow propitious, but expected, result can be explained by the contribution of the coaxial geometry of the optical design, the FOV, and the diameter of the receiving telescope. One concern regarding the use of total column is that sometimes the lidar profile could not reach the lower stratosphere and exhibits a partial column. A calculation performed on the vertically averaged water vapor data of the CFH sondes launched during the MORGANE (Maïdo ObservatoRy Gas and Aerosol NDACC Experiment) campaign (Reunion
- Island, May 2015), shows that, up to 5 km, the cumulated water vapor represents 90% of the total column; above 10 km, 99% of the whole column is contained. Thus, it appears sufficient to use IWV total columns to calibrate the Lidar1200 water vapor profiles when the range of the profile is higher than 10 km.

#### 2.3.3 Description of the methodology

Usually a calibration coefficient is the ratio between a reference instrument, here it is the GNSS IWV, and the uncalibrated lidar IWV data. In this section, the hourly calibration coefficient, named "hourly coefficient" represents the ratio between the GNSS IWV calculated on 1-hour and the lidar IWV integrated on the same temporal extent. The averages of the hourly

coefficients of one night that will be named "nightly coefficients". In the next subsections, the "calibration coefficient" will correspond to the coefficient calculated with the methodology described below, it represents the coefficient used to calibrate the Lidar1200 water vapor dataset for supplying the national and international networks. If no instrumental change occurs, the calibration coefficient is supposed to be almost constant.

- To determine the calibration coefficient of the Lidar1200's water vapor profiles, the ratios between the GNSS and lidar IWV data are compared. The temporal coincidence between the datasets consists of taking the hourly GNSS IWV and integrating the lidar data on a 1-hour window around the hourly mean GNSS. The principle can be illustrated as followed: if the GNSS measurement corresponds to 18:00 LT, the lidar data are integrated between 17:30 and 18:25 LT (with a minimum of 45 minutes). The time series of the nightly coefficients (Fig. 2) has been used in order to identify the periods with a quasi-
- constant ratio. The nightly coefficient varies in time, probably due to the fact that the integration methods are different. Even if the GNSS and the lidar are collocated, they do not measure exactly the same volume of water vapor (in space and in time). Nevertheless, periods of quasi-stationarity of the nightly coefficient can be defined inside periods in which the lidar was considered as instrumentally stable. The average nightly coefficient of each quasi-stationary period is considered as the calibration coefficient of the associated profiles. Nevertheless, for the near-real time treatment, the profile is calibrated with
- the former validated coefficient. It remains necessary to check if there are instrumental changes in order to detect possible alteration of the coefficient, and to recalculate it if required to compile the final dataset. At the beginning of each night of measurements, a systematic lamp measurement is made in order to detect potential sudden instrumental changes due to non-identified causes. The lamp measurement allows to look at the changes that occur only on the reception part of the system. The lamp measurement consists in the emission of a white light on the telescope which lightens the sensors. An acquisition is
- launched, the information extracted is "white noise". This noise is averaged all over the altitude range and for all the channels. Finally, the lamp measurement value corresponds to the ratio of the signal of the Raman H<sub>2</sub>O channel and the Raman N<sub>2</sub> channel. This value is independent from the altitude and is impacted by the laser power, the losses due to the reception optics and the efficiency of the sensors. Some of the instrumental changes affecting the calibration coefficient are known. The logbook can be checked to reduce non-automatic detections of instrumental changes. With hindsight on the dataset, the
- identification of the periods with a stable nightly/hourly coefficient allows to overcome the problem of automatic detection of instrumental change. Once the periods are identified, the calibration coefficients are validated and, thus, the data can be calibrated and used for geophysical purposes.

Figure 2 shows the time series of the nightly coefficients and the 10 associated quasi-stationary periods, with lamp measurements superimposed. An analysis of the lamp measurements leads to the identification of one very short quasi-

- stationary period associated with changes of the calibration coefficient: P03 to P04 in June 2014 (change of the interference filter of the 407 nm channel). All remaining quasi-stationary periods involving a new calibration coefficient have been identified thanks to the logbook: realignments of the optics of the lidar and interventions on the lasers. The 2-years database has been calibrated (Table 1) with the exception of the third period (P03) which corresponds to a training session of the technicians and scientists on the alignment of the water vapor lidar, which was part of a training exercise dedicated to
- preparation for the MORGANE campaign.

# 2.3.4 Comparison with other calibration devices

The previous subsection has shown that the GNSS technique provides a suitable methodology to calibrate the Lidar1200. To evaluate how suitable is this methodology, other generally used calibration techniques have been compared, in particular the use of radiosoundings. The first exercise consists of comparing the GNSS-based calibration coefficients with those derived

from the other techniques. The most common technique to calibrate the lidar raw profiles is the use of a coincident (temporally and when possible geographically) radiosounding. Dionisi et al. (2015) showed encouraging results for the use of GNSS IWV-based on a first comparison with the use of Vaisala RS92 for calibrating the profiles during MALICCA-1 campaign. To

evaluate the robustness of the GNSS calibration, the method described above needed to be further investigated based on more comparisons. The MORGANE campaign involving CFH, Vaisala RS92 and RS41 and Modem M10 sondes launched from the Maïdo Observatory concomitantly with GNSS and lidar measurements provided an interesting opportunity. The comparison method was based on a daily calibration of the lidar profiles using a GNSS or a sonde reference profile, i.e. the ratio of the

- reference profiles with the uncalibrated lidar profile of the night at an altitude between 3 and 4 km. This range of altitude has been chosen because the difference between the reference and the uncalibrated profiles was the weakest on average. This method produces a time series of daily calibration coefficients for the duration of the campaign for each type of sonde and for the GNSS (Fig. 3). Only the results for the nights when the water vapor lidar was operating and when a radiosounding has been performed simultaneously are shown. Note that the period of study includes the prolongation of the MORGANE
- campaign in June justified by the need of supplementary ozone measurements. The mean GNSS calibration coefficient on the period is 157, the one by CFH is 170, 149 for the RS92 and RS41, and 135 for the M10 (Fig. 3). The dry bias of the Vaisala and Modem sondes is known (Miloshevich et al., 2006, Bock et al., 2013). The calibration coefficient characterized by GNSS calculated with the routine methodology of the calibration of the Lidar1200 is 155 (± 32). All the coefficients derived from the different sondes are included in the standard deviation interval of the routine based calculation (123-187). The mean value of
- the coefficient derived from GNSS sits between the one derived from the capacitive sensor technique and the one derived from the CFH technique (Fig. 3). Thus, it is confirmed here that the GNSS technique is as suitable as radiosoundings for the calibration of the water vapor profiles of the Lidar1200.

#### 2.3.5 Limits of the methodology

The accuracy of the GNSS calibration method is reduced during the dry season. The total column calculated using the vertical profile measured by the CFH sonde and those obtained by GNSS show an average difference of 2 mm. If the CFH is considered as the reference instrument, the uncertainty on the GNSS IWV appeared to be of around 1-2 mm, which is in agreement with the values of accuracy in the literature. Figure 4 represents the distribution of the measurements of the GNSS IWV with season at the Maïdo Observatory. During the summer and the autumn, the distributions are monomodal and imply the largest IWV (30-35 mm). During the winter and the spring, most common IWV are around 3-5 mm. The austral winter has the driest

atmosphere, the IWV can be lower than 3 mm. Thus, the uncertainty on the calibration becomes too large during that season in particular. This effect is specific to the subtropical climate regime in Reunion Island area, where very dry conditions are observed above 5 km altitude from July to September. If any change occurs, the continuity is insured by the lamp measurements. Calibration would be improved by launching a few radiosoundings in winter from the Maïdo Observatory to confirm, or determine, the calibration coefficient of the water vapor profiles during these dry conditions.

#### 30 2.3.6 Systematic error or bias on the measurement

Calibration coefficients are calculated on the average of the individual nightly coefficients over the quasi-stationary periods. The uncertainty on the calibration coefficient of each period can be estimated by the standard deviation of the nightly coefficients of the associated quasi-stationary period. The uncertainty of the calibration coefficient can be estimated by the standard deviation of the nightly coefficients. If the calibration is considered as stationary and only due to random fluctuations,

the uncertainty on the calibration coefficient of each period is mainly due to the term corresponding to the standard deviation divided by the square of the number of nightly calibration coefficients. The uncertainty of the calibration coefficient varies between 1.6 and 3.6% between 2013 and 2015 (Table 1). The mean standard error on the calibration for the whole dataset is 2.7%.

An original methodology for Raman lidar calibration has been designed with GNSS measurements. The very low 40 values of the standard calibration uncertainty guarantee a high reliability of the coefficient calibration. The methodology minimizes the calibration uncertainty and its use has been successfully tested on the 2-years dataset of the Lidar1200. The