# Peer review of "A Raman lidar at Maïdo Observatory (Reunion Island) to measure water vapor in the troposphere and lower stratosphere: calibration and validation"

_Atmospheric Measurement Techniques, 2017_

## Short Comment (SC1) · 25 Apr 2017

General comments: The paper presents a novel instrument (LIDAR1200) with very interesting performance characteristics, new is probably also the concept of routinely calibrating such an instrument with GNSS IWV. The consideration about the accuracy of this technique are however somewhat questionable. The paper states that with the presented technique water vapor measurements in the UT/LS region can be routinely performed. Given the potentially large uncertainty of these measurements the usefulness of these observations are not fully demonstrated. There are some significant

doubts about the estimation of the uncertainty of the calibration technique. As pointed out above, not enough evidence is provided that the new technique provides sufficient accuracy to study variability and long term trends in UT/LS water vapor concentrations. The opportunity of accessing all data of the instrument should provide the opportunity to reproduce results. However, the description of calibration and validation techniques is not sufficiently clear to allow full reproduction of results, for example, how the values in table 1 are calculated is not well described.

Quality of the Presentation: A good overview of related work and results is provided in the introductory part. The number of figures and tables could be reduced. Table 2 and 3 could be omitted, e.g the vertical resolution has already been shown in Fig. 1; the campaigns in fig. 5. Fig 13 and 14 are also not particularly useful.

More detailed Comments:

Introduction: Necessity and quantitative requirements (uncertainty, long term stability) could be better motivated: who needs this data for what (which modells)? Which uncertainty would be required to be able to detect long term stratospheric water vapor changes? Section 1 A formular demonstration how WVMR is calculated from signals could be shown. Section 2.1 It seems somewhat questionable that with a telescope of 1.2 m Diameter (and 4 m focal length?) a signal from 15m distance can be detected, normally direct beams from that low an altitude will be obscured by the secondary mirror. It is likely that the observed signal is created from multiple scattering. Some considerations of the validity of the retrieval method under these circumstances should be added. Section 2.3.1 The data indicate an uncertainty in the GNNS IWV measurements of 8 -20 %. The GNSS calibrated lidar measurements can hardly be better than that. Section 2.3.4 I am not quite convinced by this analysis. It would have been better to consider the radiosonde data as reference and compare to the GNSS calibrated lidar results. Section 2.3.5 This suggests that under dry conditions (IWV=3mm) the calibration error can be as high as 66%) (2mm accuracy in the GNSS IWV) Section 2.3.6 I disagree to the statement :" the uncertainty on the calibration coefficient of each period is mainly due to the term corresponding to the standard deviation divided by the square of the number of nightly calibration coefficients." This calculation yields the uncertainty about the average of nightly coefficients, but this does not correspond to the uncertainty of the calibration, which is not based on this average. If I understand correctly, the lidar is re-calibrated every night using the GNSS, thus the uncertainty is not based on the average of the nightly coefficients but on the uncertainty of each individual calibration GNSS nightly value. This will be mainly determined by the uncertainty of the GNSS IWV measurement itself which, as pointed out earlier is not better than 8%, plus random errors. Figure 3 shows the differences between the GNSS calibration and other calibrations based on radiosonde, these differences are quite large and illustrate the uncertainty of the GNSS calibration technique. Section 3.4. It is unclear whether the lidar has been calibrated by the GNSS IWV or by the CFH data. The conclusion is not quite true, between 14 and 16 km the two instruments are not in agreement.

Typos: In line 33-34 a full sentence is repeated.

---

## Referee Comment (RC2) · D. N. Whiteman (Referee) · 24 May 2017

Review of "A Raman lidar at Maïdo Observatory (Reunion Island) to measure water vapor in the troposphere and lower stratosphere: calibration and validation" Vérèmes et al. Submitted by David N. Whiteman

**General Comments**

It is exciting and very impressive to have a Raman lidar in the Southern Hemisphere dedicated to research on stratosphere-troposphere exchange processes and the long-term monitoring of water vapor trends in the upper troposphere and lower stratosphere. The commitment of the French and European research institutions to these tasks should be greatly applauded and these efforts needs to be documented in the refereed literature. I find, however, that in its current form this manuscript needs too much revision to be publishable. I recommend that the authors carefully consider the comments below and revise the paper accordingly and resubmit. General comments are given first then more specific detailed comments follow.

1. The goal of this paper is to demonstrate that the Maido water vapor lidar is ready to supply quality data for various scientific investigations such as process studies and trend detection. The data quality needed to address these two types of studies, however, is very different. A general discussion is needed that deals with the lidar measurement uncertainty and measurement and calibration requirements for addressing different scientific studies. Then the authors can assess how well this new instrument meets these measurement needs. Documents that pertain to this type of discussion include:
   1. For discussions of measurement requirements for addressing particular scientific studies
      1. GCOS-134. Appendix 1 gives water vapor calibration stability requirement of 0.3% per decade for revealing water vapor trends.
      2. GCOS-171 The GCOS Reference Upper Air Network Guide (2013) in particular see sections 4, 7
      3. Whiteman, D. N., K. C. Vermeesch, L. D. Oman, and E. C. Weatherhead (2011), The relative importance of random error and observation frequency in detecting trends in upper tropospheric water vapor, J. Geophys. Res., 116, D21118, doi:10.1029/2011JD016610.
   2. For discussions of measurement uncertainty and calculating the total uncertainty budget of the lidar water vapor mixing ratio data product, I recommend starting with a fully detailed version of the lidar equation so that a correlation between the equation and the uncertainty terms evaluated can be described. Also, authors are advised to consult the following for assessing total uncertainty budget for the water vapor mixing ratio calculation
      1. Immler et al., Atmos. Meas. Tech., 3, 1217-1231, 2010 http://www.atmos-meas-tech.net/3/1217/2010/doi:10.5194/amt-3-1217-2010
      2. GCOS-171 Section 3.
      3. Whiteman, D. N., Cadirola, M., Venable, D., Calhoun, M., Miloshevich, L., Vermeesch, K., Twigg, L., Dirisu, A., Hurst, D., Hall, E., Jordan, A., and Vömel, H.: Correction technique for Raman water vapor lidar signal-dependent bias and suitability for water vapor trend monitoring in the upper troposphere, Atmos. Meas. Tech., 5, 2893-2916, doi:10.5194/amt-5-2893-2012, 2012. See Appendix A3.
2. After establishing the data requirements and lidar total uncertainty as requested above, the authors need to carefully justify that calibrating the Raman water vapor lidar data product

with respect to GPS IWV is sufficient to meet the measurement requirements. As mentioned in the specific comments, it appears that the uncertainty in the GPS measurements themselves (before considering the uncertainty in transferring the GPS calibration to the lidar) may be 10-20% for as much as half the measurement periods studied. The authors plan to use lamp measurements to carry the calibration through periods of higher uncertainty in the GPS data so that would imply using the lamp measurements to carry calibrations forward for perhaps half the year. But the lamp measurements do not seem to be sensitive to some of the large changes in calibration that occur (Fig 2). These details need to be carefully considered.
3. For easier editing in the future, I suggest using line numbers that carry through the entire manuscript (as opposed to starting at 1 on each page) for easier editing.

**Specific Comments**

Introduction

1. I find this first important paragraph to be rather disjointed in its logical flow due to the large number of disparate topics that the authors attempt to cover in one paragraph. The range of topics introduced in this one paragraph would demand several paragraphs to smooth the discussion for the reader. Also some statements are unclear. I suggest a very significant re-write of this material. Here are examples
   1. the second sentence discusses "long term series" while the third sentence talks about selection criteria (not yet discussed) being important for international networks (not yet discussed).
   2. The fifth sentence introduces the need for meta data which is a fully different topic than what precedes it.
   3. Lines 38-39 introduce uncertainty, algorithms and calibration which are large topics unto themselves.
   4. Line 39 states "This rigor..." but what does "This" refer to? There is no rigor that is previously described that "This" refers to.
   5. Page 2, line 1. The sentence "One of the challenging ECV to measure is water vapor mainly in the upper troposphere and lower stratosphere (GCOS, 2003)" could be more clearly stated as "Water vapor is a challenging ECV to measure in the upper troposphere and lower stratosphere (GCOS)".
   6. Page 2, line 2. The sentence "Water vapor is the main greenhouse gas" really does not fit in logically at this point. Such an assertion needs to come much earlier in the discussion as support for why the current effort is being undertaken.
   7. Page 2, line 4. please provide references also for the "transport and dynamical processes from eddies to synoptic scale events" portion of this sentence.
   8. Lines 5-6 discuss spatial variability of water vapor without regard to whether they are referring to upper troposphere or the lower stratosphere. The statements they make are much more appropriate for the troposphere than the lower stratosphere. This is an important distinction for the authors to make as they present different temporal and spatial averaging schemes for the lidar data processing later in the paper.
   9. Line 7 introduces the concept of measuring trends in water vapor and what is needed to do so. But it is already published that monitoring trends of water vapor greatly depends on whether you are referring to upper troposphere or lower stratospheric trends. That distinction is not discussed here or elsewhere and surely needs to be. See reference below

1. Whiteman, D. N., K. C. Vermeesch, L. D. Oman, and E. C. Weatherhead (2011), The relative importance of random error and observation frequency in detecting trends in upper tropospheric water vapor, J. Geophys. Res., 116, D21118, doi:10.1029/2011JD016610.
   10. The remaining sentences in the paragraph introduce NDACC and GRUAN but make no mention of trend detection in reference to either of these networks. Instead, the discussion shifts to UT/LS exchange (for NDACC) and "characterization" for GRUAN.
2. Line 19. "Among the radiosondes, the hygrometers are the most efficient". This sentence refers to a mixture of technologies and makes an inaccurate statement. Radiosondes measure temperature, pressure, RH and winds typically while hygrometers measure water vapor alone so hygrometers should not be considered "among the radiosondes". And I suspect that the authors may be assuming that "hygrometer" refers to those instruments that measure water vapor using the chilled mirror technique, but that is not correct. Hygrometer is a more generic term referring to any instrument that measures the water vapor content. Also, what does it mean for a hygrometer to be the "most efficient"? I suspect the authors mean to refer to accuracy in some way instead.
3. Line 20 contains a misstatement about CFH claimed accuracy from the 2007 Voemel paper. The 4% accuracy figure relates to the tropical lower troposphere not the tropical lower stratosphere.
4. Line 21 "They are shown to be good in the UT/LS " is not a very scientific statement. What does "good" mean? How is it quantified?
5. Line 22. Statement is made "Nevertheless, the CFH are rarely launched on a routine basis at these stations mainly because of their cost." What stations are referred to here? There are no stations discussed elsewhere in the paragraph.
6. Line 24 states that "the MLS (Microwave Limb Sounder) is very accurate (Read et al., 2007) and even the most efficient in the lower stratosphere". Again I ask the question of what efficient refers to here since this claim is not referenced. But also, are the authors aware of the recent divergence in the lower stratosphere between MLS and frostpoint hygrometer that Hurst et al. have documented?
   1. Atmos. Meas. Tech., 9, 4447–4457, 2016 www.atmos-meas-tech.net/9/4447/2016/ doi:10.5194/amt-9-4447-2016
7. Line 41. Authors reference Sherlock et al. to support a statement of the importance of calibration stability. But the Sherlock paper presented an independent calibration technique which did not refer to another measurement of water vapor to calibrate the Raman lidar. Following line 41, the authors only discuss dependent calibration activities, i.e. ones where the Raman lidar calibration is derived from another measurement of water vapor. Authors should include a discussion of both dependent and independent calibration techniques, such as documented in Venable et al which presented an alternative independent calibration technique which needs to be referenced in the discussion.
   1. DD Venable et al. Appl Opt 50 (23), 4622-4632. 2011 Aug 10
8. Page 3, line 1. Statement is made that all the Raman lidars of NDACC use Vaisala sondes to calibrate their database. This is not the case as the independent technique described in Venable et al. has been implemented and used in the ALVICE Raman lidar, a member of NDACC.
9. Line 14. Statement is made "Some critical points have been addressed in the upgrade, including fluorescence, power and parallax effects, in order to optimize the configuration of the system (Hoareau et al., 2012; Sherlock et al., 1999b)". I suggest some additional text to explain to the reader at least something about what these critical upgrades are. Since they are so important, the reader should have more explanation about them here without having to read the referenced papers.

10. Line 31. Statements are made "It is noteworthy that this Maïdo Raman water vapor lidar (called hereafter "Lidar1200") was recently provisionally affiliated within the NDACC. The conclusive affiliation occurs when absence of fluorescence and a stable calibration method are both demonstrated using validation campaigns involving frost-point hygrometer measurements." It would be very informative to discuss what the calibration stability requirements are to meet the NDACC goals of process studies and trend monitoring. These two types of studies have very different calibration stability requirements and those should be detailed.

11. Page 4, line 13. Statement is made "A spectrometer is used directly after this telescope to separate the Raman and Rayleigh signals." Traditionally the term spectrometer is used to refer to a grating spectrometer, which can be used to perform the task at hand. Here the authors are using a standard combination of beamsplitters and interference filters, which only becomes apparent later in the discussion. I suggest using a different term here such as "wavelength separation package" or some such term and quickly state that it consists of beamsplitters and interference filters.

12. Page 4, line 16. Statement is made "The overlap factor is identical for both channels". Perhaps the two functions are quite similar, but surely they are not identical. Some quantification of how similar they are and what the authors did to quantify it is needed here. Also realize that what we call the channel overlap function contains any position dependent optical efficiency variations in the beam splitters, interference filters and pmts. So it is highly likely that the overlap functions will have at least some small differences from one channel to the next.

13. Line 24, section 2.2.1. The authors present Eq 1 as the "total absolute error" of the water vapor measurement, yet the formula presented is not appropriate to account for systematic uncertainties of which there are certainly several in the water vapor calibration. The authors are referred to an earlier publication which attempted to present the full uncertainty budget of the water vapor mixing ratio calculated by a Raman lidar involved in the MOHAVE 2009 campaign. See Appendix A3. It's particularly important to note that both the uncertainty of the calibration source and the uncertainty in transferring that calibration to the Raman lidar mixing ratio measurement are separate and important sources of uncertainty, both of which are systematic and not random. There are other uncertainties not considered by the authors as well.
    1. Whiteman, D. N., Cadirola, M., Venable, D., Calhoun, M., Miloshevich, L., Vermeesch, K., Twigg, L., Dirisu, A., Hurst, D., Hall, E., Jordan, A., and Vömel, H.: Correction technique for Raman water vapor lidar signal-dependent bias and suitability for water vapor trend monitoring in the upper troposphere, Atmos. Meas. Tech., 5, 2893-2916, doi:10.5194/amt-5-2893-2012, 2012.

14. In the discussion of uncertainty, the terms "uncertainty" and "error" are both used. It would be useful to clarify what the difference is that the authors are making. More traditional might be to use the terms "random uncertainty" and "systematic uncertainty" and not use the term "error".

15. Page 5, line 3. The statement is made "Indeed, the effect of aerosols on Raman channels in the UV is low." It's not fully clear what the authors mean by this statement but aerosol attenuation of the UV Raman signals under discussion is quite significant under turbid conditions. Errors in quantifying the aerosol extinction profile result in systematic uncertainties in the transmission profile. See the following publications:
    1. Whiteman, David N., Examination of the traditional Raman lidar technique. II. Evaluating the ratios for water vapor and aerosols, Applied Optics, 42, No. 15, 2593-2608 (2003). See Fig 8.
    2. Veselovskii et al., Atmos. Meas. Tech., 8, 4111–4122, 2015 www.atmos-meas-tech.net/8/4111/2015/ doi:10.5194/amt-8-4111-2015 Figs 1, 2 etc.

16. Line 11. It is notable and somewhat confusing in a calibration paper, such as this, which claims to be presenting the "absolute error" of the Raman lidar water vapor mixing ratio data product

that, referring to the reference GPS IWV product, "The complete evaluation of the uncertainty will be further detailed in a future publication." Knowledge of the total uncertainty of the GPS calibration source is needed to quantify the total uncertainty of the lidar calibration which depends on the GPS. Authors should consider this point and revise their discussion appropriately.

17. Line 14. Statement is made "Data are smoothed with a filter using the Blackman coefficients:...". This statement sounds like it is meant to follow earlier ones that have introduced what kind of filter is used and other details about it. So without that earlier material, this sentence is confusing to the reader. Please detail what filter is used and what the Blackman coefficients are.

18. Line 30. "the signal travelling between a GPS satellite (altitude of 20,200 km) and a ground-based receiver is delayed by atmospheric constituents (dry air, and water vapor) ..." Other atmospheric constituents such as clouds, hydrometeors, aerosols also influence the propagation of the microwave signals associated with GPS. The temperature profile of the atmosphere is also important for determining the signal delay. Please revise text.

19. Page 7, line 3. "If no instrumental change occurs, the calibration coefficient is supposed to be almost constant." The variability of the calibration constant is a subject of this investigation. The calibration value has certain variability which the authors are in the process of quantifying. I suggest removing statements like these and replace them with statements that provide quantities with uncertainties that specify variability based on their data analysis.

20. Paragraph starting line 5. This paragraph reads as if the lidar is calibrated each night using the nightly calibration coefficient. Figure 2 seems to contradict that concept so the discussion is confusing. There is a significant detail that bares mentioning in how frequently the lidar is re-calibrated. As mentioned earlier, the uncertainty due to transferring the calibration of the GPS to the Raman lidar entails a systematic uncertainty. Previous field campaign research shows that this systematic uncertainty is typically 2-5% depending on the particular experiment. Given that time series of lower stratospheric water vapor are desired to be stable at better than 0.1 ppm level per year (~2%) (GCOS requirement), the systematic uncertainty associated with the transfer of a calibration from another instrument to the Raman lidar water vapor mixing ratio data product, by itself without considering any other sources of uncertainty, can be sufficient to make the time series of greatly reduced value or even useless for lower stratospheric trend detection. This statement is true assuming, as is usually the case, that the calibration coefficient is determined only infrequently (every several months or perhaps once per year) based on ensemble comparisons with radiosondes and then carried forward using lamp-based measurements. The way to address this weakness of the dependent calibration technique is to perform the dependent calibration frequently enough such that the uncertainty associated with the calibration transfer process becomes part of the random uncertainty budget instead of being part of the systematic uncertainty budget. This is the technique that the ARM Raman lidar has followed since its inception by using the technique of a running calibration with respect to microwave radiometer and one that the authors might be able to use here if quality GPS calibrations are available on a daily basis. While lower stratospheric water vapor measurements do not benefit from an increase in random uncertainty, at least random uncertainty can be reduced by making additional measurements. Introducing 2-5% systematic uncertainties in a time series by, for example annual changes in the calibration coefficient would prevent trends at the 1% per year (which is a common estimate of the magnitude of the water vapor trend in the UT/LS) to be determined. The authors are referred to the following publication which states the importance of converting sources of systematic uncertainty to random uncertainty when possible. The point should be made in this paragraph that the daily determination of the

calibration coefficient by comparison with GPS turns the systematic uncertainty associated with the transfer of calibration from GPS into a component of the random uncertainty budget.

1.  Whiteman, D. N., K. C. Vermeesch, L. D. Oman, and E. C. Weatherhead (2011), The relative importance of random error and observation frequency in detecting trends in upper tropospheric water vapor, J. Geophys. Res., 116, D21118, doi:10.1029/2011JD016610.

21. Lines 7-8. reference is made to 1-hr time windows for lidar integration but the example times cover 55 minutes. Please reconcile.

22. Figure 2. There are several very significant changes in calibration based on the GPS measurements that seem not to be identified by the lamp measurements. Is this the case? Why don't the lamp measurements identify these large changes in calibration coefficient? What is the explanation for this? Authors should be aware of several "failure modes" of the lamp-based technique that are described in the reference below. Is one of those failure modes in play here?

1.  Whiteman, D. N., D. Venable, E. Landulfo, Comments on "Accuracy of Raman lidar water vapor calibration and its applicability to long-term measurements", Applied Optics Vol. 50, Iss. 15, pp. 2170–2176 (2011)

23. Table 1 presents the calibrations used during the quasi-stationary periods. Please define more clearly the difference between "absolute" and "relative" error and, again, the term "uncertainty" is preferred over "error". If the relative error that the authors refer to is the uncertainty of the transfer of the calibration from the GPS to the lidar, then this needs to be acknowledged as a systematic uncertainty in the time series that is introduced each time the re-calibration is done.

24. Page 8, lines 10-11. Please provide the standard deviation of the derived calibration coefficients. Statement is made that the Vaisala sondes have a known dry bias and the authors use this statement in reference to the RS41 in addition to the RS92. The RS41 was not studied in the Miloshevich and Bock works cited, however, so it is not a proper reference. The apparent good agreement shown by the authors between RS41 and RS92 in the 3-4 km range and the recent measurement campaigns of the RS41 showing very good performance of the new instrument would tend to indicate that both RS92 and RS41 sensors were performing very well during the campaign in the required 3-4 km range. This contrasts with the authors claim of dry bias.

25.  Line 13. Lidar1200 calibration using the "routine method of calibration" is given as 155+/-32. What is the "routine method" and how do the authors reconcile a 21% calibration uncertainty figure given here with the much smaller calibration uncertainty values shown in Table 1?

26. Line 17. "Thus, it is confirmed here that the GNSS technique is as suitable as radiosoundings for the calibration of the water vapor profiles of the Lidar1200. " None of the calibrations shown here seems to offer as stable a calibration value as shown in Table 1. Also, were the lamp measurements useful in quantify the same large changes in calibration coefficient shown in Figure 3?

27. Section 2.3.5. Authors state that IWV comparisons with CFH show an uncertainty in the GPS IWV measurements of 1-2 mm. From Figure 4 it appears that as much as 50% of the time, the IWV at the site is 10 mm or less. Therefore, from the authors estimates, the GPS IWV uncertainty would seem to be 10-20% during approximately half of the measurement periods. Is this calibration uncertainty acceptable for use as the calibration source for measurements to be used within NDACC? The authors state that the lamp measurements can be used to carry the calibration forward during these dry periods, but the lamp measurement results shown in Figure 2 do not seem to show sensitivity to some of the large calibration variations that occur so can the authors really rely on the lamp to carry the calibration forward? And getting back to one of the main questions, authors need to discuss what the calibration accuracy and stability requirements are for water vapor data to be useful for both process studies and trend detection. The former has considerably more stringent accuracy and stability requirements than the latter and that would be useful for the authors to detail here since, given the above considerations,

there could be broad skepticism among readers about these measurements being suitable for trend studies.

28. Section 2.3.6. The authors discuss here what I have referred to as the uncertainty in the transfer of the calibration coefficient from the external measurement of water vapor to the Raman lidar water vapor profile. The authors have correctly identified, by the title of this section, that this is a source of systematic uncertainty in the total error budget and so cannot be propagated as they have shown in Eq. 1. Please reconcile.

29. Line 34 "If the calibration is considered as stationary and only due to random fluctuations, the uncertainty on the calibration coefficient of each period is mainly due to the term corresponding to the standard deviation divided by the square of the number of nightly calibration coefficients."  The authors need to be clear that the value they are considering here is the uncertainty in the transfer of the calibration from GPS to lidar and does not consider the uncertainty in the GPS calibration itself. The variability in this calibration transfer coefficient is surely influenced by, perhaps dominated by, atmospheric variability since the GPS is sampling a large volume whereas the lidar is sampling just the atmosphere directly overhead.  Also, the atmospheric conditions are different each time the calibration transfer is done. So the assumption that the variation in the calibration transfer coefficient is only due to random fluctuations is not in general satisfied. Thus it is not correct to divide by the square root (not square as stated by authors) number of samples in calculating the uncertainty in this transfer of calibration. The more conservative way to perform this calculation, so as to specify an upper bound to the uncertainty in the transfer of the calibration, is to simply use the standard deviation (and not standard error) as the uncertainty for the transfer of calibration and not divide by the Sqrt[N] term. The real uncertainty is likely somewhere in between the standard deviation and the standard error, but is surely larger than the standard error that the authors have used.

    1. It should be noted that the uncertainty of the calibration of the external IWV measurement iself needs also to be factored into the total uncertainty budget of the calibration. That also is a source of systematic and not random uncertainty.

30. Page 9. Line 25. "With regard to relative humidity, it is recognized by many that the CFH sondes are among the most accurate especially in the UT/LS (Vömel et al., 2007)." There are two dominant cryogenic frostpoint hygrometer instruments in the world currently, the CFH and the NOAA FPH. They are considered comparable in performance. So the statement cited is not correct.

    1. Atmos. Meas. Tech., 9, 4295–4310, 2016 www.atmos-meas-tech.net/9/4295/2016/ doi:10.5194/amt-9-4295-2016

31. Section 3.3. Given the highly variable signal to noise of the lidar versus the relatively constant one of frostpoint, the technique described for comparing the two instruments is quite reasonable. One thing that is puzzling, though, is why such a high power, large aperture lidar system requires such long averaging time (48 hours) to produce a quality profile extending beyond 20km. Previous research shows Raman water vapor lidar measurements extending to these altitudes with lower laser power (16 vs 24W), smaller telescope (0.6 vs 1.2m) and shorter averaging time (9 vs 48 hr). In considering the relative signal to noise of these two measurement examples, the difference must be in the noise term instead of the signal term. For the higher performance measurements, the noise was reduced by use of a 0.25 mrad field of view, 0.25 nm interference filter and thermo-electrically cooled water vapor PMT. The authors may want to consider whether further optimizations of the noise term would reduce the averaging time required to probe the lower stratosphere. It should be noted that in the MOHAVE measurements cited below no indication of fluorescence was present and agreement with CFH and climatology was very good. See Fig 13 from the reference below.

1. Whiteman, David N., Kurt Rush, Scott Rabenhorst, Wayne Welch, Martin Cadirola, Gerry McIntire, Felicita Russo, Mariana Adam, Demetrius Venable and Rasheen Connell, Igor Veselovskii, Ricardo Forno, Bernd Mielke and Bernhard Stein, Thierry Leblanc and Stuart McDermid, Holger Vömel, Airborne and Ground-based measurements using a High-Performance Raman Lidar, doi:10.1175/2010JTECHA1391.1  (2010).

32. Line 22. Referring to the comparisons shown in Figure 6, statement is made "No positive or negative bias appears." By contrast the figures seem to indicate clear biases between lidar and CFH in the 3-5 km range that sometimes exceed 20%. Please reconcile. The later statements in the paragraph relating to the case of 19 May may offer an explanation for the biases, but this can be checked by performing the comparisons excluding the 19 May case. But certainly the quoted statement is incorrect and needs to be changed.

33. Line 38. "There is no obvious reason to explain this bias". Agreed. It could be an indication of PMT signal induced noise with just the right decay constant, but that is just speculation.

34. Page 11. Line 2. "To conclude, the Lidar1200 and the CFH profiles are in a good agreement in the whole region of the troposphere sampled by Lidar1200, and the MORGANE campaign profiles have been validated by the CFH sondes up to 22 km asl." This statement is not consistent with the significant biases in the altitude ranges of 3-5 and 14-16 km. Please reconcile.

35. Figure 7. See earlier discussion about errors and uncertainties. There are contributions not considered here and I do not believe that the authors have properly calculated the total calibration uncertainty.

36. Section 4.1. There is much discussion here of how long an averaging time is required to reach what altitude. But such discussion needs to be based on what the measurement requirements are for certain types of studies and, as mentioned before, that material is lacking. Authors need to add an early important section about what random and total uncertainty are acceptable for the types of analyses (e.g. process studies, trend detection) they want to do with the data from this instrument.

37. Section 4.2 Some introduction to this section is needed that explains what the authors mean by optimal performance. As it stands it is not really clear what the intent of this section is.

38. Figure 8. I am not sure I see the value of including this figure. I think the authors can discuss these data without showing this figure.

39. Page 12. Line 17. "The profile of 24 September 2015 integrated on the all-night measurements reaches approximately 18 to 19 km altitude with an uncertainty of the order of magnitude of 35% (4-5 ppmv, Fig. 8)."  I doubt that 35% uncertainty corresponds to 4-5 ppm in the lower stratosphere based on climatology or MLS. A value closer to 2 ppmv seems more reasonable. It might help to show other measurements/climatology for comparison.

40. Line 27. "The upper limit is the altitude where the lidar uncertainty corresponds to twice the variability of the water vapor in the lower stratosphere." Different values of "variability" are considered but they do not seem to be the result of an analysis. So variability needs to be defined here and how it is calculated must be presented.

41. Figure 9. The first three plots show unreasonable values in the upper several km. The authors are advised to overlay MLS (even with the recent divergence issue discussed in Hurst et al. it will be useful since the divergence is measured in tenths of ppmv) or climatological data for illustration of this. Only the 4th plot seems to have a reasonable behavior throughout its range. An overlay of other LS data onto these plots would make this point.

42. Figure 10. Please see earlier discussions concerning sources of uncertainty and their calculation.

43. Section 4.3. Authors discuss the influence of varying averaging times on revealing fine structures in the atmosphere. These are good illustrations of the need for algorithms that automatically make best use of the information content of the data. Other methods of processing

the water vapor data such as adaptive or optimal estimation algorithms have been presented before that permit these kind of structures to be optimally revealed in an operational way. See relevant publications below:

1. See Section A.4 of previously mentioned Whiteman, D. N. et al. : Correction technique for Raman water vapor lidar signal-dependent bias and suitability for water vapor trend monitoring in the upper troposphere, Atmos. Meas. Tech., 5, 2893-2916, doi:10.5194/amt-5-2893-2012, 2012.
2. Sica RJ, Haefele A., Retrieval of water vapor mixing ratio from a multiple channel Raman-scatter lidar using an optimal estimation method., Appl Opt. 2016 Feb 1;55(4):763-77. doi: 10.1364/AO.55.000763.

44. Section 5.2. How does the seasonal cycle captured by the lidar measurements compare with space-borne measurements (e.g. MLS) or climatology? Please discuss.
45. Section 6. "The spatio-temporal variability of the water vapor is not well documented through direct observations." The near global measurements of space-borne sensors such as MLS and others contradict this statement. Please reconcile.
46. Acknowledgments. "The lidar data used in this publication were obtained as part of the Network for the Detection of Atmospheric Composition Change (NDACC) and level 2 product as daily vertical water vapor profiles will be publicly available through the NDACC portal (http://www.ndacc.org) and the French atmospheric data portal (http://www.pole-ether.fr/). " Authors provide 2 sites where water vapor data may be downloaded, but neither seems to have data from this system. How can one access the measurements?

---

## Author Comment (AC1) · 18 Sep 2017

Dear Reviewer,

We would like to thank you for your insightful and helpful comments and suggestions. Please find in the supplement file a point by point response along with the new version of the manuscript.

Best regards.

[Figure]

Please also note the supplement to this comment:
https://www.atmos-meas-tech-discuss.net/amt-2017-32/amt-2017-32-AC1-
supplement.zip

---

## Author Comment (AC2) · 18 Sep 2017

Dear Reviewer,

We would like to thank you for your insightful and helpful comments and suggestions. Please find in the attached supplement a point by point response along with revised manuscript.

Best regards.

[Figure]

Please also note the supplement to this comment:
https://www.atmos-meas-tech-discuss.net/amt-2017-32/amt-2017-32-AC2-
supplement.zip